# Emerging Trends in Magnetic Resonance Fingerprinting for Quantitative Biomedical Imaging Applications: A Review

**DOI:** 10.3390/bioengineering11030236

**Published:** 2024-02-28

**Authors:** Anmol Monga, Dilbag Singh, Hector L. de Moura, Xiaoxia Zhang, Marcelo V. W. Zibetti, Ravinder R. Regatte

**Affiliations:** Center of Biomedical Imaging, Department of Radiology, New York University Grossman School of Medicine, New York, NY 10016, USA; dilbag.singh@nyulangone.org (D.S.); hector.lisedemoura@nyulangone.org (H.L.d.M.); xiaoxia.zhang@nyulangone.org (X.Z.); marcelo.wustzibetti@nyulangone.org (M.V.W.Z.)

**Keywords:** magnetic resonance imaging, magnetic resonance fingerprinting, quantitative imaging, medical imaging, image reconstruction, deep learning

## Abstract

Magnetic resonance imaging (MRI) stands as a vital medical imaging technique, renowned for its ability to offer high-resolution images of the human body with remarkable soft-tissue contrast. This enables healthcare professionals to gain valuable insights into various aspects of the human body, including morphology, structural integrity, and physiological processes. Quantitative imaging provides compositional measurements of the human body, but, currently, either it takes a long scan time or is limited to low spatial resolutions. Undersampled k-space data acquisitions have significantly helped to reduce MRI scan time, while compressed sensing (CS) and deep learning (DL) reconstructions have mitigated the associated undersampling artifacts. Alternatively, magnetic resonance fingerprinting (MRF) provides an efficient and versatile framework to acquire and quantify multiple tissue properties simultaneously from a single fast MRI scan. The MRF framework involves four key aspects: (1) pulse sequence design; (2) rapid (undersampled) data acquisition; (3) encoding of tissue properties in MR signal evolutions or fingerprints; and (4) simultaneous recovery of multiple quantitative spatial maps. This paper provides an extensive literature review of the MRF framework, addressing the trends associated with these four key aspects. There are specific challenges in MRF for all ranges of magnetic field strengths and all body parts, which can present opportunities for further investigation. We aim to review the best practices in each key aspect of MRF, as well as for different applications, such as cardiac, brain, and musculoskeletal imaging, among others. A comprehensive review of these applications will enable us to assess future trends and their implications for the translation of MRF into these biomedical imaging applications.

## 1. Introduction

Magnetic Resonance (MR) techniques, such as Magnetic Resonance Imaging (MRI) and Magnetic Resonance Spectroscopy (MRS), are extensively used in medicine and biology. MR techniques can identify variations in tissue properties. Conventional weighted MR images are more qualitative than quantitative and thus are usually referred to as qualitative MRI images. The clinical community has favored qualitative images due to their fast acquisition and good anatomical contrast and the familiarity with reading weighted images among trained radiologists. On the other hand, quantitative imaging provides an objective, more specific, and standardized measurement of tissue properties. The standardized measurement makes the quantitative images reproducible across scanners, vendors, and time. Quantitative imaging lends itself well to automated diagnostics [1] and radiomics [2,3]. It is also suitable for monitoring disease progression using MRI [4]. 

In traditional quantitative MRI approaches [1,5,6], multiple images with changes in a single parameter are usually acquired, such as multiple T1-weighted images. Then, a single quantitative map is estimated by applying relaxometry measurements to these images. This single quantitative parametric map is usually sensitive to more than one pathology at a time, which restricts its specificity. There is a need to acquire more than one quantitative parameter to improve the specificity of the quantitative evaluation. However, this implies that other MR parametric maps will have to be acquired, one by one, which takes a long time when traditional quantitative MRI is used. This increases scan costs, and patient discomfort makes the acquisition extremely susceptible to misalignment. Accelerated MRI using k-space undersampling, such as Compressed Sensing (CS), can still be used in traditional quantitative MRI, but only modest acceleration factors can be obtained because parametric maps are still being acquired one by one. 

Unlike traditional quantitative imaging methods, magnetic resonance fingerprinting (MRF) introduces a joint framework capable of acquiring and reconstructing multiple parametric maps quickly, simultaneously, and with perfect alignment.

### 1.1. Overview of MRF

In an MRF pulse sequence, the sequence parameters, such as Repetition Time (TR) and Flip Angle (FA), are dynamically varied throughout the scan, as illustrated in Figure 1A, creating varied temporal signal patterns depending on the type of tissue. At each excitation pulse, extremely undersampled k-space data are acquired. The acquisition is designed in such a way that the k-space trajectories of each pulse do not entirely overlap with each other, ensuring the acquisition of diverse k-space information, but also such that it consistently acquires small portions of the center of the k-space. The raw acquired data usually form a 2D dataset, where one dimension represents the k-space position and the other reflects the time in which the patterns evolved. Subsequently, an undersampled image series is reconstructed from the acquired k-space data, as depicted Figure 1B, providing insights into the evolution of the signals in the voxels over time. The tissue properties are extracted from the measured undersampled signal. In the MRF framework, this is achieved by comparing the measured signal against a set of simulated ideal signal evolutions (their fingerprints) with known relaxation times ($T_{1}$, $T_{2}$, $T_{1\rho }$, etc.) and other properties (like B0 and B1 fields) that could belong to that particular voxel. The collection of fingerprints is also called a dictionary, as depicted in Figure 1C. The true signal evolution is matched with the simulated signal evolutions in the dictionary. The best-matched component of this dictionary is then assumed to be the actual signal evolution of a particular voxel, as shown in Figure 1D. Through this process, MR properties, such as $T_{1}$ and $T_{2}$ relaxation times, are estimated, offering a more comprehensive and quantitative characterization of the imaged tissues, as shown in Figure 1E. The MRF technique thus enables simultaneous sensitivity to multiple MR parameters, providing an efficient approach to quantitative MRI. 

### 1.2. Related Works

The first paper on MRF was published in 2013 [8]. Since then, there has been a considerable increase in publications on MRF and corresponding review papers that summarize the developments in the field. In [9], a review of technical developments in MRF until 2019 is provided. The authors discussed the acquisition, dictionary generation, reconstruction, and validation of several MRF sequences. In [10], the authors highlighted the challenges that need to be overcome to make MRF viable in clinical settings and provided recommendations for the same. In [11], an update of [10] is provided, aiming to highlight the technical developments in MRF relating to the optimization of acquisition, reconstruction, and machine learning. In [12], the authors provided a systematic review, primarily focusing on the implementation of MRF in clinical settings and on the challenges that need to be addressed, such as improving the standardization of MRF and its implications for radiologists. There have been several review papers targeting specific clinical domains, such as cardiology, radiotherapy, and cancer. In [13], the authors discussed the technical details of the cardiac MRF and initial clinical validation of cardiac MRF. In [14], the authors discussed the technical and potential clinical application of MRF in the characterization of cardiomyopathies, tissue characterization in the left atrium and right ventricle, post-cardiac transplantation assessment, reduction in contrast material, pre-procedural planning for electrophysiology interventions, and imaging of patients with implanted devices. In [15], the authors discussed technical developments at the intersection of artificial intelligence and MRF for cardiac imaging. In [16], the authors discussed challenges and recent developments in integrating MRF into the radiotherapy pipeline. In [7], the authors summarized the latest findings and technological developments for the use of MRF in cancer management and suggested possible future implications of MRF in characterizing tumor heterogeneity and response assessment.

### 1.3. Contributions

The main contributions of the review paper are two-fold: (a) to provide a comprehensive discussion of emerging trends in the technical aspects of MRF, specifically focusing on data acquisition methodology, dictionary generation techniques, and advancements in parametric map reconstruction; (b) to explore the diverse applications of MRF across various domains, illustrating the evolution and progress observed over the years. By synthesizing information from both technical advancements and application domains, this review paper seeks to offer an up-to-date and insightful overview of the current state-of-the-art and future directions of MRF applications. In Table 1, we can see that this paper provides an extensive and complete discussion of technical trends in MRF. Further, we extensively discuss the application of deep learning models for MRF reconstruction and sequence optimization. The limitation of the paper is that it does not discuss the challenges and trends in the application of MRF in clinical settings. These specific aspects of MRF have been extensively discussed in [10,12].

## 2. Emerging Trends in MRF Pulse Sequences

MR pulse sequence selection is crucial in MRF experiments, as it plays a core role in both exciting and sensing the various tissue properties. The initial MRF demonstration [8] employed an Inversion Recovery (IR) balanced Steady-State Free Precession (bSSFP) sequence with spiral k-space trajectories, which is a sequence with a good Signal-to-Noise Ratio (SNR) and sensitiveness to T_1_ and T_2_ processes. Though a wise initial choice for the initial MRF demonstration, the IR-bSSFP is susceptible to B_0_ field inhomogeneities, creating inhomogeneity broadening that ultimately leads to mismatches between the expected signal evolution and the measured one [18]. In response to the limitations of bSSFP, spoiled gradient echo (GRE) sequences like Fast Imaging with Steady State Precession (FISP) [19] and Fast Low-Angle SHot (FLASH) have gained prominence in MRF acquisition. These sequences, featuring gradient and RF spoiling mechanisms, partially mitigate the effects of B_0_ inhomogeneity by making the transverse magnetic field incoherent. However, this advantage comes with a trade-off of a reduced SNR. 

To address challenges associated with the transient state of MRF with Steady-State Free Precession (SSFP), in [20], the authors introduced innovative conditions involving changes in flip angle and phase. These conditions aim to adiabatically maintain spin magnetization in the same direction as the effective magnetic field, minimizing the complex component of the spin magnetization. This strategic adjustment effectively reduces the impact of inhomogeneity broadening. The MR acquisition is also susceptible to B_1+_ heterogeneities. In [21], the authors proposed a plug-and-play (PnP) MRF approach to solve B_1+_ heterogeneities by encoding it in the MRF dictionary, allowing the estimation of parametric maps and local B_1+_ fields.

Quantitative parameters other than $T_{1}$ and $T_{2}$ can be estimated using MRF reconstruction. In [22], the authors proposed a novel flow MRF sequence that acquires 3D flow velocity along with $T_{1}$ and $T_{2}$ maps. A tri-directional bipolar gradient is applied between the spin excitation RF pulse and the time of acquisition to make the sequence sensitive to 3D flow velocity. The bipolar gradient pulse and flip angle are constantly changed in each repetition to encode the variation in $T_{1}$, $T_{2}$, and flow velocity. In [23], the authors proposed the use of RF excitation with quadratic phase increments along with flip angles, allowing the MRF sequence to encode off-resonance effects and $T_{2}^{*}$. In [24], the authors demonstrated the feasibility of a multi-dimensional MRF sequence to jointly quantify the relaxation, such as $T_{1}$, $T_{2}$, and diffusion of water molecules. A FISP-based MRF sequence provided the most accurate diffusion quantification compared to the FLASH-based MRF sequence. Similar to flow MRF, spherical and linear bipolar gradients are used as a preparation for spoiled GRE sequences to quantify 2D diffusion. The MRF sequence is sensitive to $T_{1\rho }$ magnetic relaxation time using multiple spin-lock preparations [25] while varying the FAs and TRs. 

In [26], a gradient echo technique, called Quick Echo Splitting (QUEST), is proposed for MRF acquisition. It departs from the gradient recalled echo (i.e., SSFP, FISP, and FLASH) paradigm usually used in MRF acquisition. In QUEST-MRF, more echoes are acquired than the number of RF pulses; this leads to a paradigm in which less RF energy is required in QUEST-MRF compared to gradient recalled echo sequences. The QUEST-MRF leads to a lower specific absorption rate (SAR) and can be used for ultra-high-field scanners.

## 3. Emerging Trends in MRF Sequence Optimization

MRF offers significant flexibility in designing the acquisition process. On top of choosing the pulse sequence, the user can control all the pulse sequence parameters, such as FAs and TRs. While the initial MRF demonstration [8] suggested that this could be randomly chosen, the researchers quickly realized that randomness is not necessarily the best approach, since the output of MRF acquisition is a signal evolution contaminated by under sampling artifacts, noise, and phase inhomogeneity. All these effects can add randomness to the measured signal evolution, leading to ambiguity and erroneous pattern matching. Thus, optimization of the acquisition parameters is more effective in encoding the tissue properties, while, at the same time, maintaining distinct signal evolutions and high SNR.

To optimize the MRF sequence parameters, two steps are necessary: Define metrics to measure the encoding capabilities of MRF acquisition strategies.Choose the algorithm that will be used to optimize the sequence parameters.

To encode tissue properties in the MRF signal, the chosen optimization metric should ensure that the MRF signal evolutions for closely related $T_{1}$ and $T_{2}$ values are in proximity to each other. However, the metric should also be designed to push signals with disparate $T_{1}$ and $T_{2}$ values further apart from each other. In [27], the authors define an objective function that optimizes the encoding ability of MRF signals. Equation (1) represents the optimization problem, where θ corresponds to the acquisition parameters and $D_{\theta }$ is the dictionary comprising signal evolution for different tissues obtained using θ. The objective of the optimization problem is to improve the ability to distinguish tissue properties, such as $T_{1}$ and $T_{2}$, by increasing the linear independence of dictionary columns.
(1)$$\hat{\theta }=\underset{\theta }{\mathrm{argmin}}{\left\|I-D_{\theta }^{T}D_{\theta }\right\|}_{F}$$

The authors compared several optimization algorithms, concluding that interior point methods are the ones most suited for this problem. 

In [28], the authors proposed metrics to measure the local and global encoding properties of MRF signal evolutions. Inner product and mean square error are used to measure the proximity of MRF signal evolutions. For proximal tissue properties, the inner product needs to be close to the unity, while the mean square error needs to be as small as possible. In [29], the authors proposed an equation that models aliasing and noise introduced into the signals during MRF reconstruction. By optimizing the proposed model, the effects of the aliasing and the noise are reduced in the measured parametric maps. In [30], the optimal acquisition parameters are calculated using total error (noise and aliasing) as a cost function to be minimized. Different from the previous works, this new model considers the k-space trajectory and the image-reconstruction algorithm. Minimization is carried out using simulated annealing [31] and Monte Carlo algorithms [32]. 

The Cramer–Rao Lower Bound (CRLB) measures a lower bound for the variance of an unbiased estimator. In [33], the authors proposed the optimization of the CRLB concerning the MRF acquisition parameters. The authors minimized the CRLB using Sequential Quadratic Programming (SQP). Optimizing CRLB is a computationally expensive task. Because of this, in [34], the authors proposed an automatic differentiation approach that is claimed to be more computationally efficient than SQP or heuristic optimization. In [33], it was observed that, in practice, effective FA and TR values are smoothly varying, with very few abrupt changes. Consequently, in [35], the authors proposed to represent the FA and TR trains by a reduced set of b-splice coefficients, reducing the number of components to be optimized. This compressed representation makes the CLRB optimization more computationally efficient.

In recent years, a multitude of research papers have employed deep learning techniques to enhance the reconstruction of MRI and MRF. Some studies have focused on leveraging deep learning models to optimize the acquisition parameters, particularly in making MRF acquisition sensitive to Magnetization Transfer (MT) effects. In [36], the authors assume a two-compartment model for each voxel involving free water and semi-solid states, with magnetization transferring between these two states. To optimize the acquisition parameters (e.g., saturation time, TRs, and FAs), the paper proposes a feedforward deep learning solution. This approach not only optimizes the model weights but also fine-tunes the acquisition parameters. In [37], the authors introduce an acquisition schedule optimization strategy for a three-compartment voxel model, employing the so-called MRF Deep RecOnstruction NEtwork (DRONE). The DRONE network is trained to map simulated signal evolutions to tissue properties. To further refine the acquisition parameters, a surrogate network was proposed. This surrogate network maps acquisition parameters to reconstruction loss, and the acquisition parameters are adjusted to minimize this loss. The overall objective is to enhance the efficiency of the acquisition schedule and improve the accuracy of tissue property estimation through the integration of deep learning methodologies.

## 4. MRF Dictionary Generation

MRF acquisitions are simulated for different tissue properties and stored in a dictionary. The dictionary contains the templates that are matched with each MRF acquisition to reconstruct parametric maps. In this section, we discuss the tools used to simulate the MRF sequences to produce these templates. The bSSFP-MRF sequences with variable FAs and TRs can be simulated using a well-established Bloch equation formalism, as shown in [8]. The dictionary is created with simulated signal evolutions for a range of $T_{1}$ and $T_{2}$ values. Since the bSSFP signal is sensitive to off-resonance effects, the dictionary comprises signals for a range of off-resonance effects. In GRE sequences, simulating spoiling using the Bloch equation formalism is challenging. To simulate spoiling for a single voxel, several signal evolutions with varying phase shifts need to be calculated, making it computationally expensive. Hence, the Extended Phase Graph (EPG) framework [38] has been proposed, which can simulate magnetization spoiling using a shift operation. EPG frameworks can be extended to model systems with magnetization transfer. In [39], the authors proposed a model (EPG-X) for a coupled two-compartment system, with each compartment having a separate phase graph that exchanges magnetization during signal evolution. The EPG-X framework was able to model signal evolution with a better fit for bovine serum albumin (MT effects) phantoms compared to the EPG framework. In quantitative MR, the non-ideal slice profile of the RF pulse creates the effect of different FAs across slices, and the actual signal deviates considerably from the desired signal. MRF acquisition can be made robust to these distortions in the acquisition by modelling these slice distortion effects. In [40], the authors extend the EPG formalism to include distortions introduced by the slice profile of the RF pulse. The EPG formalism is further extended to support simulating anisotropic diffusion imaging in the 1D direction in the work of [41]. In [42], a new phase graph formalism is introduced where 3D gradients can be simulated, proving useful for modeling MRF acquisition with diffusion gradients in three directions. Ref. [43] proposed a hybrid Bloch–EPG formalism that can predict effects on acquisition due to the slice profile of the RF pulse, off-resonance, spoiling moment, microscopic dephasing, and echo time. It can model both SSFP and spoiled GRE sequences interchangeably. 

## 5. Emerging Trends in MRF Reconstruction

MRF reconstruction aims to convert an acquired k-space signal into quantitative maps, like $T_{1}$ and $T_{2}$, expressing the tissue compositional information. Figure 2 illustrates the important MRF reconstruction pipelines that have been proposed over the years. Figure 2a illustrates the general framework of image reconstruction. All MRF image reconstruction can be broken down into two fundamental steps: 1. reconstruction of a temporal image from a temporal k-space; 2. mapping the parametric maps from the temporal image. The nature of these fundamental steps helps us classify MRF reconstruction, as shown subsequently. 

### 5.1. Dictionary-Based MRF Reconstructions

As illustrated by Figure 2b, there are two steps involved in dictionary-based reconstruction: (1) mapping the undersampled k-space data back to the image sequence domain and (2) matching the measured signal evolution of each voxel with pre-computed entries of the dictionary. In [8], the authors reconstructed the image space from k-space using Non-Uniform Fast Fourier Transformation (NUFFT) followed by dictionary matching in the image space to map tissue properties. The authors demonstrated that dictionary matching is robust to undersampling artifacts and motion artifacts. The dictionary should encompass the tissue properties corresponding to the age, gender, and pathology of the subjects to be imaged and the region of interest. However, large-sized dictionaries are computationally inefficient and memory-demanding. Hence, to build a dictionary with an affordable size, a proper grid with the right number of discrete values of the tissue properties must be defined, which is not always possible without introducing discretization errors. To overcome the computational inefficiencies of large-sized dictionaries, based on the assumption of low rankness of a dictionary, it can be compressed, as proposed in [46]. Another approach to improve efficiency is to reduce the number of comparisons, using a systematic search. This approach was demonstrated in [47,48]. In [48], the authors break down the dictionary into groups, with each group having a specific signature. A hierarchal matching process is performed, where the MRF acquisition is matched against a group signature to obtain the best-matched group. A higher-resolved parametric map is obtained by matching signal evolution only with the elements of the best-matched group. In [47], the authors propose an efficient and fast search mechanism called Dictionary Generation and Search (DGS). It dramatically speeds up the matching algorithm without losing accuracy.

The distance metrics used to match the signal evolution with the dictionary have an important role in the accuracy and robustness of the dictionary matching step. There are multiple distance metrics used in the MRF literature. The most popular among them is the inner product of normalized signal evolution with a normalized dictionary. The index corresponding to the maximum inner product points to the closest tissue properties. Inner product as a distance matrix is first referenced in [8,19]. In [8], the authors interestingly proposed a potential approach to resolve multiple components from a single voxel in the supplementary documents. The authors proposed that acquired MRF signal evolution could be modeled as a weighted sum of signal evolutions in the dictionary. The weights of the individual components could be calculated by optimizing a least-square difference between the signal evolution and the weighted combination of dictionary signal evolution. A multi-compartment model for dictionary matching is further discussed in Section 5.4. 

In [49], the authors proposed a matching strategy based on learning an optimal distance metric. The distance metric used was based on the Mahalanobis norm. The Relevant Component Analysis (RCA) algorithm [50] was used to learn the optimal distance metric. It was shown to improve the dictionary matching process compared to the inner product metric. 

### 5.2. Dictionary- and Model-Based MRF Reconstructions

The previous MRF framework does not use any special operator, other than the inverse NUFFT, to map the data from the k-space to the image domain. This results in each image being highly corrupted by undersampling artifacts. In contrast, model-based reconstructions, as illustrated by Figure 2c, are used to improve the mapping from the k-space to the image domain. In [51], a model-based CS method called Bloch response recovery via Iterated Projection (BLIP) was proposed. The BLIP approach reconstructs the image iteratively while projecting the image onto the Bloch equations manifold. The acquired MRF acquisition exhibits redundancy in the temporal domain. Hence, in [52], the authors proposed a CS model-based MRF image reconstruction called Fingerprinting with LOw-Rank constraint (FLOR). In this model, a low-rank regularizer is used to model the temporal redundancy. In [53], a similar CS reconstruction approach called Model-Based Iterative Reconstruction (MBIR) was proposed which incorporates a low-rank image model while estimating the tissue parametric maps directly from k-space. In [54], the authors assumed the low-rank image model for MRF reconstruction and learns the low-dimensional subspace from the dictionary. During the model-based reconstruction, an image evolution projected onto a lower-dimensional subspace is estimated. The dictionary matching is performed on the low-dimensional image evolution to estimate the parametric maps. In [45,55], CS model-based reconstruction approaches were proposed which exploit low-rank image models and impose constraints on the optimization problem, such that the signal evolutions in the dictionary and the measured signal evolution are not far apart. It enforces the low-rank constraint and projection onto the dictionary, thereby reducing artifacts.

In [56,57], the authors introduced a dictionary-based Bayesian learning approach to characterize tissue properties. Dictionary-based Bayesian learning approaches can quantify multiple tissue properties within the same voxel. In [56], the authors showed that for vascular imaging, the Bayesian learning approach is robust to high levels of noise and more accurate than conventional dictionary matching. 

### 5.3. Deep Learning-Based MRF Reconstructions

In recent years, there has been a growing trend towards the use of deep learning methods in image reconstruction in MRI and quantitative MRI. Dictionary matching is both computationally and memory-inefficient. Deep learning models enable dictionary-less MRF reconstruction. Deep learning models require fewer computational and memory resources compared to dictionary-based MRF reconstruction. Figure 2d,e highlights the configuration of pipelines for deep learning-based MRF reconstruction. Figure 2e illustrates an MRF reconstruction pipeline where deep learning models are used to map signal evolutions to parametric maps. Figure 2d illustrates an unrolled deep learning network that maps k-spaces directly to the parametric maps.

In [58], the authors proposed a feedforward neural network model to map signal evolution in a single voxel to the parametric map. The multi-dimensional signal evolution (image space) is reconstructed from k-space data using sliding window inverse NUFFT. Similarly, in [59], the authors proposed a feedforward neural network to map $T_{1}$ and $T_{2}$ values from signal evolution in a single voxel. The multi-dimensional signal evolution is reconstructed from k-space using inverse NUFFT (without the sliding window strategy). The signal evolution is filtered using k-SVD to make the feedforward network robust to aliasing and noise. The model was tested with FISP, bSSFP, IR-FISP, and IR-bSSFP. IR-FISP was the most accurate in estimating the tissue properties. In [60], deep learning (DL) models like the Multi-Layer Perceptron (MLP) and a Recursive Neural Network (RNN) were trained to calculate T_1_ and T_2_ values for signal evolution in each voxel; the proposed network was tested on a 7T preclinical scanner on a rat brain phantom. This work demonstrates that proposed DL models such as RNN and MLP are more accurate at quantification when compared to other convolution-based neural networks (UNET, CNN, and CED) and conventional dictionary-matching methods. In [61], a novel deep learning approach is proposed, where a channel-wise attention mechanism is used to enhance the focus on informative channels to reconstruct quantitative parametric maps from reconstructed MRF signal evolutions. In [62], the authors proposed a residual UNET with a channel attention module to quantify $T_{1}$ and $T_{2}$ values. Both these methods improved the accuracy of $T_{1}$ and $T_{2}$ estimation from MRF signal evolution error compared to dictionary matching and convolutional deep learning models. In the above references, it can be seen that deep learning models are particularly useful to map image evolution to parametric maps. In [63], the authors proposed a generative deep learning model as a prior model to enhance the accuracy of k-space to image space reconstruction. This model improves the accuracy of the image space reconstruction in in vivo experiments in comparison to low-rank prior, as shown in [54]. In recent years, [44,64] have demonstrated that unrolling deep learning networks can be used directly to map the parametric maps from k-space acquisition. In [44], the approach is inspired by the following optimization problem:(2)$$\hat{x}=\underset{x\in B}{\min}\left\|Ax-k\right\|+\lambda T\left(x\right),$$
where x is the reconstructed image; k is the measured k-space data; and B is the set of all images in which the signal evolutions of each voxel satisfy the Bloch equation (such as those in a dictionary). $T\left(x\right)$ is a tensor low-rank regularization term with a weighting parameter, λ. $\left\|Ax-k\right\|$ is the data-consistency norm. Equation (2) is minimized to solve x from k while keeping x consistent with Bloch equations. The matrix, A, comprises a cascade of transforms (NUFFT; autocalibration transform for parallel imaging; undersampling transform). The solution for Equation (2) can be approximated by alternating between solving the following three subproblems: (1) enforcing consistency between k-space and the image evolution reconstructed, which is called the data-consistency module (DC); (2) enforcing low rankness of the reconstructed image evolution, which is called the learned decomposition module (CP); and (3) enforcing the consistency of the reconstructed image evolution with the Bloch equation. The module is called the Bloch response manifold module (BM). All these steps are formulated in a deep learning model [44]. In [44], parametric maps were estimated using a deep learning model, with the output of BM acting as model input. The paper demonstrates increased accuracy and reconstruction speed compared to [51,52,53] for brain imaging. In [64], the authors proposed an unrolled network that combines the CP and BM into a single step and uses a deep learning model to approximate the step. The DC step uses a conventional linear function.

### 5.4. Other MRF Reconstruction Frameworks

Several ancillary limitations arise in the MRF reconstruction process. There can be motion artifacts, off-resonance artifacts, B_1_ inhomogeneities, and partial volume effects. All these factors can affect the MRF reconstruction. In this section, we discuss the MRF reconstruction methodologies that explicitly solve these problems. 

To model the partial volume effect, the signal evolution is modeled as a weighted combination of sub-voxel signal evolution. The distribution of the tissue properties of sub-voxels is obtained using an inverse problem. In [57], a Bayesian approach, using Gaussian models for tissue properties, is proposed. In [65], the authors propose an inverse problem to estimate the weight of each tissue in the sub-voxel with a weighted low-rank constraint. In [66], an extension of the idea from [65] is proposed, using a joint sparsity constraint. An efficient optimization algorithm called Sparsity Promoting Iterative Joint NNLS (SPIJN) is used to solve tissue properties in the sub-voxel. In [67], the authors compare the partial volume MRF obtained by a pseudo-inverse formulation and dictionary-based method. The authors demonstrate that dictionary-based partial volume MRF has lower sensitivity to noise. In cardiac imaging, there is a periodic motion caused by heartbeats that can affect the quantification of tissue properties. In [68], a deep learning approach is proposed to estimate the $T_{1}$ and $T_{2}$ mapping in cardiac MRF. The deep learning model was trained on data with simulated periodic motion and noise. In [69,70], the authors propose MRF reconstruction frameworks accounting for rigid motion and non-rigid motion, respectively. The bSSFP sequence is susceptible to B_0_ inhomogeneity, often resulting in erroneous patterns in the signal evolution that can affect the estimation of $T_{1}$ and $T_{2}$ values and MRF reconstruction. In [71], the authors propose a model-based MRF reconstruction that incorporates a low-dimensional learned non-linear manifold using a deep autoencoder. It improves the estimation of $T_{1}$ and $T_{2}$ parametric maps in bSSFP-based MRF sequences. In [72], off-resonance artifacts are modeled as a point spread function over the true magnetization. The authors use conventional MRF acquisition to estimate $T_{1},\;T_{2}$, and $∆f_{off}$. The correction for $T_{1}$ and $T_{2}$ is estimated from linear regression, with the point spread function (estimated using $∆f_{off}$) acting as a transform. The method showed improved reconstruction compared to the conventional dictionary matching.

To make MRF clinically relevant for radiologists, we need to incorporate mechanisms to reconstruct fully sampled contrast-weighted images like $T_{1}$-weighted, $T_{2}$-weighted, and FLAIR in the MRF reconstruction pipeline. In [73,74], the authors propose synthetic contrast generation from MRF data using deep learning methods.

## 6. Emerging Trends in MRF Applications

In this section, we discuss recent trends in MRF applications, classifying the recent publications into cardiac, brain, musculoskeletal, and abdominal imaging. The papers are also classified according to tasks, such as fat–water separation and radiotherapy, as seen in Table 2. 

### 6.1. Cardiac Imaging

Cardiovascular magnetic resonance (CMR) imaging is an important non-invasive method to characterize myocardial tissue. The characterization of tissue properties can be achieved by tissue-specific quantitative parameters like $T_{1}$, $T_{1\rho }$, $T_{2}$, $T_{2}^{*}$, and extracellular volume (ECV). MRF can be modified to consider the peculiarities [14] of CMR imaging and reconstruct multiple cardiac tissue parameters simultaneously. The heart beats with a periodic rhythm (cardiac rhythm). This rhythm varies from person to person based on several physiological parameters. The cardiac rhythm adds complexity to CMR imaging. A CMR imaging framework has to acquire the k-space within the duration of a cardiac cycle; CMR imaging is susceptible to motion artifacts due to cardiac rhythms. Figure 3 demonstrates a general cardiac MRF (cMRF) parametric map reconstruction pipeline. The first application of MRF for cardiac imaging was demonstrated in [101] with an MRI scanner. The ECG triggered cMRF-FISP-based acquisition with a breath hold every 16 heartbeats and was used to reconstruct 2D $T_{1}$ and $T_{2}$ parametric maps. Because of the randomness in heartbeats, every acquisition needs to have a customized dictionary. $T_{1}$ and $T_{2}$ parametric maps showed a bias of 1 ms and −2.6 ms, respectively, compared to conventional cardiac quantitative MRI images. Three-dimensional coverage of MRF is required to make a precise estimation of tissue properties of the myocardial muscles. To improve the coverage of cMRF, the authors proposed a simultaneous multi-slice cMRF in [102], where multiple slices of the images are acquired at the same time. A 3D approach for cMRF was proposed in [75]. The authors proposed a FISP-based sequence comprising an inversion recovery pulse and a $T_{2}$ preparation pulse with free-breathing acquisition. Three-dimensional acquisition is susceptible to motion artifacts; hence, a motion-corrected Low-Rank Inversion (LRI)-based High-Dimensionality undersampled Patch-based Reconstruction (HD-PROST) is used for reconstructing the MRF images. The parametric maps are reconstructed using template matching. The 3D-cMRF acquires each slice 7 min faster than clinical standards for $T_{1}$ and $T_{2}$ quantitative maps. In [77], the authors proposed a single breath-hold 2D Dixon-cMRF framework to simultaneously estimate $T_{1}$, $T_{2}$, $T_{2}^{*}$, and fat fraction (FF). In [103], 2D single breath-hold cMRF acquisition was tested for a 1.5 T scanner for repeatability of $T_{1}$ and $T_{2}$ parametric maps, and the test also compared it against parametric maps obtained from conventional clinical best practices. In [78], the cMRF acquisition proposed in [103] was extended to $T_{1}$, $T_{2}$, and $T_{1\rho }$ parametric maps for myocardial tissue with a 1.5 T scanner. The reconstructed parametric maps show a high correlation with spin-echo MRI and conventional clinical parametric maps. Cine magnetic resonance imaging is an MRI acquisition method in which the dynamic motion of the heart is captured during a heartbeat. It is used to evaluate the motion of the heart’s chambers and abnormalities in the contraction and relaxation of the heart. In [104], the authors proposed a novel cine-MRF with a single 16-heartbeat breath hold to simultaneously acquire cine images along with $T_{1}$ and $T_{2}$ parametric maps.

The long acquisition window in cMRF makes it particularly vulnerable to artifacts in the case of a subject with a high heart rate. In [105], the authors demonstrated that cMRF is less precise than conventional quantitative MRI acquisitions especially for subjects with high heart rates. The advantage of cMRF is that it allows for higher-resolution acquisition.

### 6.2. Brain Imaging

MRI is a non-invasive approach that allows for detailed visualization of the anatomy, structure, and function of the brain. Different imaging biomarkers measure different aspects of the tissues in the brain. $T_{1}$ tissue properties help to differentiate between gray and white matter, whereas $T_{2}$ tissue properties highlight the water content in the brain and aid in identifying lesions, edemas, and other abnormalities. $T_{2}^{*}$ is sensitive to blood component in the brain and can detect hemorrhagic strokes. Diffusion Tensor Imaging (DTI) allows for mapping the microstructural organization of brain tissue. Perfusion imaging allows for measuring the blood flow to the brain. MRF allows for reconstructing multiple tissue properties simultaneously. In [79], perfusion, diffusion, $T_{1}$, and $T_{2}^{*}$ were estimated in the same acquisition. In [80], the authors proposed a Chemical Exchange Saturation Transfer (CEST) MRF with EPI acquisition to quantify tissue properties in brain tumors. A tumor comprises water and semi-solids which interact with each other, to which CEST-MRF is perfectly suited. The water $T_{1}$; water $T_{2}$; and the rate of exchange between water, amide and semi-solid is measured from the MRF acquisition using a deep-learning model. In [82], the viability of MRF acquisition in detecting Parkinson’s disease in a person is demonstrated. A significant difference was seen between healthy control and subjects with Parkinson’s disease for $T_{1}$ and $T_{2}$ mapping in different subregions of the brain, as shown in Figure 4. In [83], the authors proposed an MRF Arterial Spin Labeling (ASL) scan that can measure the blood flow in the brain (brain hemodynamics). The quantitative maps measured from MRF acquisition are blood flow, Cerebral Blood Volume (CBV), and $T_{1}$. The deep learning model is used to map MRF signal evolution to tissue properties. 

In [106], the authors proposed an interleaved Echo-Planar Imaging (EPI)-MRF acquisition that can acquire high-resolution multi-slice whole-brain-coverage (resolution: 1 mm × 1 mm × 3 mm) in 3 min 36 s. The $T_{1}$, $T_{2}^{*}$, and PD quantitative maps mapped from the acquisition have high visual quality compared to 2D-MRF-EPI acquisition with a 4-fold time reduction. Similarly, in [107], a 3D coverage of the brain with a resolution of 1.2 × 1.2 × 3$\;{mm}^{3}$ was achieved using an interleaved spiral MRF acquisition in 4.6 min. An illustration of the method is shown in Figure 5.

### 6.3. Musculoskeletal Imaging

The musculoskeletal (MSK) system forms a key component of the human body. It comprises the spine, hips, knees, all-important joints, bones, and muscles in the body. $T_{1}$, $T_{2}$, $T_{1\rho }$, and FF are the key tissue properties that can indicate the health of the MSK system. In [86], the authors tested a Plug-n-Play MRF acquisition method in estimating the $T_{1}$, $T_{2}$, and T_1ρ_ of the knee’s articular cartilage (Figure 6a). The paper demonstrates the repeatability and strong correlation between conventional MRI quantification and MRF. T_1ρ_ shows significant separability between healthy and osteoarthritis (OA) subjects, as shown in Figure 6b. In [87], an ultra-short echo time MRF acquisition method was proposed that is sensitive to ultrashort $T_{2}$ values. The T_2_* weighting effect is reduced in such acquisitions and is useful for estimating tissue properties in muscle, bone, ligaments, and tendons. In [89], the authors used in-phase and out-phase TE patterns along with variable flip angles to measure fat–water separation along with $T_{1}$ and $T_{2}$. An MRF reconstruction was postulated which can make $T_{1}$ and $T_{2}$ less sensitive to fat, which is useful for measuring $T_{1}$ and $T_{2}$ values in the muscles and joints. In [90], an MRF acquisition method was proposed that measures FF maps, water $T_{1}$, and fat $T_{1}$. MRF acquisition framework similar to [86] was used to estimate $T_{1}$, $T_{2}$, and T_1ρ_ for lower legs [88], and hips [91,92] and inter-vertebral discs [108]. 

### 6.4. Abdomen Imaging

The abdominal region comprises several organs like the kidney, stomach, pancreas, gall bladder, and liver. The main challenge in imaging the abdominal region is that it is affected by respiratory and cardiac rhythms and hence is susceptible to motion artifacts. To compensate for the motion sensitivity, in [95], the authors proposed a pilot tone navigator to track motion in the abdomen and use the motion information to correct MRF acquisition. In [96], the authors proposed a free breathing MRF spiral acquisition method with oversampling at the center to reduce motion sensitivity. The feasibility of MRF acquisition for the liver and kidney is demonstrated in [97,98].

### 6.5. Radiotherapy, Tumors, and Cancer

Radiotherapy is a medical treatment where a directed dose of radiation is applied to destroy cancerous cells. MRI is used to monitor tissue changes, localize the cancerous cells, monitor motion in the cancerous region, classify tumors, and assess responses. MRF offers a framework to quantify tissue properties accurately and rapidly. In [7,16], reviews of recent technical and clinical developments for the application of MRF in radiotherapy are presented. In [80], the authors proposed and demonstrated the feasibility of quantitative CEST-based MRF to measure the tissue properties of a brain tumor (the $T_{1}$ and $T_{2}$ for water and fat and the exchange rate between the solid-state pool and the liquid pool). It can be very useful for radiotherapy. Ref. [99] demonstrates the feasibility of MRF in measuring $T_{1}$ and $T_{2}$ values to characterize ovarian tumors. It has been successfully implemented in a low-field MR-guided radiation therapy device, demonstrating its technical feasibility and accuracy, as shown in [109]. In [110,111], the authors demonstrated the feasibility of MRF to characterize prostate cancer tissue properties. In [110], it is shown that MRF-based $T_{1}$ and $T_{2}$ mapping along with ADC mapping from diffusion-weighted imaging (DWI) can differentiate between cancerous and non-cancerous lesions in the prostate. Cancerous lesions have lower ADC and $T_{1}$ values compared to non-cancerous lesions, as demonstrated in Figure 7. 

### 6.6. Fat–Water Separation

Due to the chemical environment, the Larmor frequencies of protons in fat and water are different. In MRI, depending on the type of acquisition, this leads to phase differences in the complex-valued voxels. By varying the echo times, we can separate the fat and water components. Sequences that use a three-point Dixon framework are commonly used to separate fat and water contributions. This separation is crucial in various clinical applications, such as musculoskeletal imaging, breast imaging, and abdominal imaging. In [89,94], two-point and three-point Dixon MRF methods are proposed, respectively. In both studies, Conjugate-Phase Reconstructions (CPRs) are used to quantify $T_{1}$ and $T_{2}$ and reduce biases due to fat and water components of tissues. Cardiac imaging is affected by the periodic heartbeat. In [77], the authors proposed a 15-heartbeat ECG triggered with a three-point Dixon MRF for the heart, which quantifies $T_{1}$, $T_{2}$, and FF for a single breath-hold exam. After every five heartbeats, an inversion pulse is triggered. In [90], the authors proposed an RF-spoiled multi-compartment GRE MRF to quantify the $T_{1}$ values in fat and water along with the FF for each voxel. The acquisition scheme is called Dictionary-Based Fat–Water separation (DBFW) and quantifies $T_{1}$ values more accurately than the Iterative Decomposition of water and fat with Echo Asymmetry and Least-squares estimation (IDEAL) and Dixon-MRF.

## 7. Discussion and Future Outlook

Throughout this paper, we have reviewed the emerging trends in MRF. In Section 2, the recent trends in MRF pulse sequences were reviewed. Despite the flexibility in the MRF framework in choosing the pulse sequence, most acquisitions use gradient-echo sequences, such as bSSFP and FISP, with IR pulses and sometimes FLASH segments. One of the reasons behind this choice is the good $T_{1}$ and $T_{2}$ sensitivity of bSSFP and FISP sequences, with $T_{1}$ sensitivity improved by IR pulses and sensitivity to B_1+_ inhomogeneity with FLASH segments. Another reason is the flexibility in choosing variable FAs and short TRs for easy control of the signal evolution with relatively fast acquisition.

After every FA, a short k-space readout is used. The selection of k-space trajectories in MRF acquisition is another key issue that needs to be resolved. Most of the k-space trajectories acquire the center of the k-space every readout and a different part of the medium and high frequencies. Radial trajectories with golden-angle increments are often used. Spiral trajectories are also useful; the non-linear trajectory of the spiral permits better coverage of the k-space at each RF pulse. Non-linear Cartesian trajectories, such as EPI are also used, covering a good portion of the k-space at each pulse; however, at the price of longer readouts. Table 3 compares the advantages and disadvantages of each trajectory. Note that machine-learned sampling patterns and trajectories [112,113,114] are already used by traditional quantitative MRI, but they have not been extended to MRF yet. For 3D MRF acquisition, the 2D k-space trajectories are usually stacked, as in stack-of-stars (for radial trajectories) and stack-of-spirals.

Selecting the pulse sequence and its k-space trajectory in MRF is only part of the problem. Researchers now know that random choices of FAs and TRs are not optimal. As seen in Section 3, the optimization of these parameters is essential for the success of MRF acquisition. The optimizations must target multiple features, better SNR, better sensitivity to different quantitative parameters and consider the effects of undersampling artifacts. Optimizations improving CRLB, as in [33,35], are a good choice regarding SNR and the sensitivity of quantitative parameters. However, most models become too complex when undersampling is included. In [115], based on the assumption of spatial effects of aliasing, a convolution filter is used to model the aliasing effects. It would be interesting to use such models in optimizing the acquisition parameters. A pertinent approach would be to run the simulation on multi-dimensional digital phantoms with aliasing and noise consideration and optimize the acquisition parameters based on the simulation, as demonstrated in [30]. There are still many open questions regarding the optimization of MRF pulse sequence parameters and data acquisition. 

By far, the vast majority of developments have regarded quantitative parameter reconstruction, as seen in Section 5. The number of papers using deep learning in MRF reconstruction has increased considerably in recent years. MRF reconstruction is time-consuming, computationally expensive, and unsuitable for online reconstruction. A deep learning model that can map parametric maps directly from k-space can help in reducing the computation and time requirements, thereby making MRF viable for online reconstruction.

MRF can also be used in both low-field and ultra-high-field MRI scanners. Low-field MRI scanners are usually cheaper; are characterized by shorter $T_{1}$, longer $T_{2}^{*}$/$T_{1\rho }$, and lower SAR; and are less susceptible to field inhomogeneity artifacts, especially when metal implants are present. On the other hand, low-field MRI acquisition faces limitations, such as low SNR and less spectral separation of water and fat, which demand longer acquisition times. In [109,116,117,118,119], the authors demonstrated the feasibility of MRF acquisition in low-field MRI scanners with magnetic fields ranging from 0.05 to 0.55 T. Ultra-high-field MRF, on the other hand, has a high SNR, which allows for higher resolution and faster acquisition. However, it is very susceptible to field inhomogeneities and higher SAR. Refs. [100,120,121] demonstrate the feasibility of 7 T MRF in clinical scans. In [122,123], the feasibility of MRF sequences was demonstrated at 9.4 T. 

For MRF to be relevant for clinical applications, it must establish its reliability and reproducibility across scanners, institutions, and vendors. In [124], using the MRF FISP sequence, an ISMRM/NIST phantom was repeatedly scanned over 34 days. The paper demonstrates that $T_{1}$ and $T_{2}$ value estimates were repeatable with a coefficient of variance (CV) < 5% over a wide range of $T_{1}$ and $T_{2}$ values. In [125], a multi-center study was conducted on the NIST/ISMRM phantom using 1.5 T and 3.0 T GE scanners and MRF-SSFP sequences. The paper demonstrated that, within a range of $T_{1}$ and $T_{2}$ values, the parametric maps showed strong repeatability (CV < 8%) and moderate reproducibility (CV < 3%). As shown in [126], a multi-center acquisition of phantoms and prostatic tissue was performed using five different 3.0 T MRI scanners (one Skyra and four Verio Siemens scanners) with different software versions (VE11C, VB19, and VB17) using FISP sequences. The intra-scanner ($T_{1}$: CV < 2%; $T_{2}$: CV <4.7%) and inter-scanner ($T_{1}$ CV < 4.9%; $T_{2}$ CV < 8.1%) variation for an MRF acquisition was low. Both $T_{1}$ and $T_{2}$ values in invivo prostatic tissue demonstrated high test–retest reliability. In the ISMRM/NIST phantom with $T_{2}$ < 30, the inter-scanner CV > 15% and the intrascanner CV > 3%. In general, the FISP MRF sequence is more accurate in measuring $T_{1}$ compared to $T_{2}$. The high CV at a lower $T_{2}$ can be explained by the relative coarseness of the dictionary at lower $T_{2}$ values. In the brain, multiple multi-center trials were conducted demonstrating repeatability, reproducibility, and reliability in parametric maps generated by MRF, as shown in [127,128,129]. These studies demonstrated that the repeatability and reproducibility of the brain vary based on the region of interest. In [128], the authors conducted multi-site repeatability and reproducibility experiments on 1.5 T and 3 T MRI scanners with 3 T scanners, showing better reproducibility and repeatability. The experiment in [127,129] indicated that MRF shows good reproducibility in gray and white matter compared to cerebrospinal fluid. Other than $T_{1}$ and $T_{2}$, MRF acquisition can measure metrics like $T_{1\rho }$, diffusion, and flow rate. To estimate these parametric maps and properly tune the MRF sequences, we need phantoms that are sensitive to these metrics over a range of values. In [130], the authors define the design requirement of a phantom for quantitative MRI and demonstrate examples of phantoms for different applications ranging from diffusion to flow phantoms. Once the perfect sequence is designed, repeatability and reproducibility play a key role in demonstrating the viability of MRF acquisition. 

## 8. Conclusions

Magnetic resonance imaging (MRI) plays a crucial role in medical imaging by providing high-resolution images with excellent soft-tissue contrast. This imaging modality offers valuable insights into the morphology, structural integrity, and physiological processes of the human body. However, quantitative imaging techniques face challenges, such as long scan times or limited spatial resolution. To address these challenges, techniques like undersampled k-space data acquisitions, compressed sensing, and deep learning reconstructions have been designed to reduce MRI scan times and mitigate undersampling artifacts. Additionally, MRF has emerged as an efficient framework for acquiring and estimating multiple tissue properties simultaneously in a single fast MR acquisition. Even though MRF is a relatively new quantitative MRI technique, its research interest has increased exponentially, and it has undergone multiple developments since its initial demonstration. Current research shows developments regarding pulse sequence structure and parameter optimization, reconstruction, and investigative steps toward clinical usage. The combination of knowledge in spin dynamics and undersampling makes MRF perhaps one of the best examples of effective usage of MRI scanners for simultaneous quantitative mapping of the human body.

This paper provides a comprehensive literature review of the MRF framework, highlighting trends and challenges associated with each aspect. However, despite the advancements, challenges persist in MRF across different magnetic field strengths and body parts, presenting opportunities for further investigation. By reviewing best practices in each aspect of MRF and its applications in areas such as the heart, brain, musculoskeletal system, and abdomen, this paper aims to assess future trends and their implications for the translation of MRF into biomedical imaging applications. Finally, by addressing current challenges and identifying future directions, we hope to pave the way for the continued advancement and adoption of MRF in clinical practice, ultimately benefiting patient care and diagnosis.

## Figures and Tables

**Figure 1 bioengineering-11-00236-f001:**
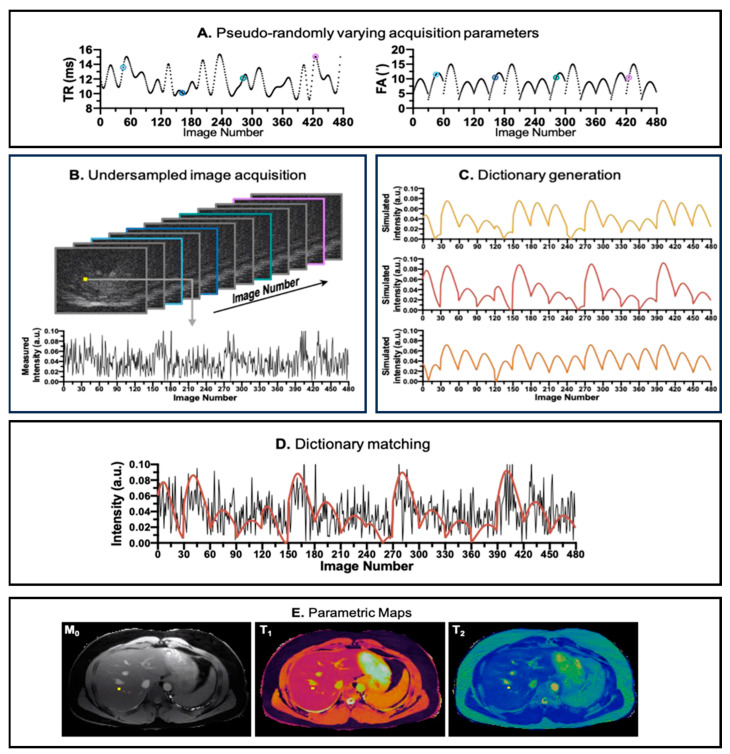
Pipeline for MRF acquisition, reconstruction, and parametric maps. (**A**) The pseudo-random repetition time (TR) and flip angle (FA) trains to introduce incoherence in acquisition. (**B**) The reconstructed image from k-space acquired in the acquisition step. In the image number dimension, the images are color coded corresponding to its TR and FA values (annotated in A). (**C**) Dictionary simulation corresponding to the tissue properties in the region of interest. (**D**) Matching between the simulated dictionary (red line) and the acquired signal evolution (black line) of a voxel. (**E**) The parametric maps ($T_{1}$, $T_{2}$, and $M_{0}$) generated after all acquired voxels are matched against the simulated dictionary. The image is derived from [7].

**Figure 2 bioengineering-11-00236-f002:**
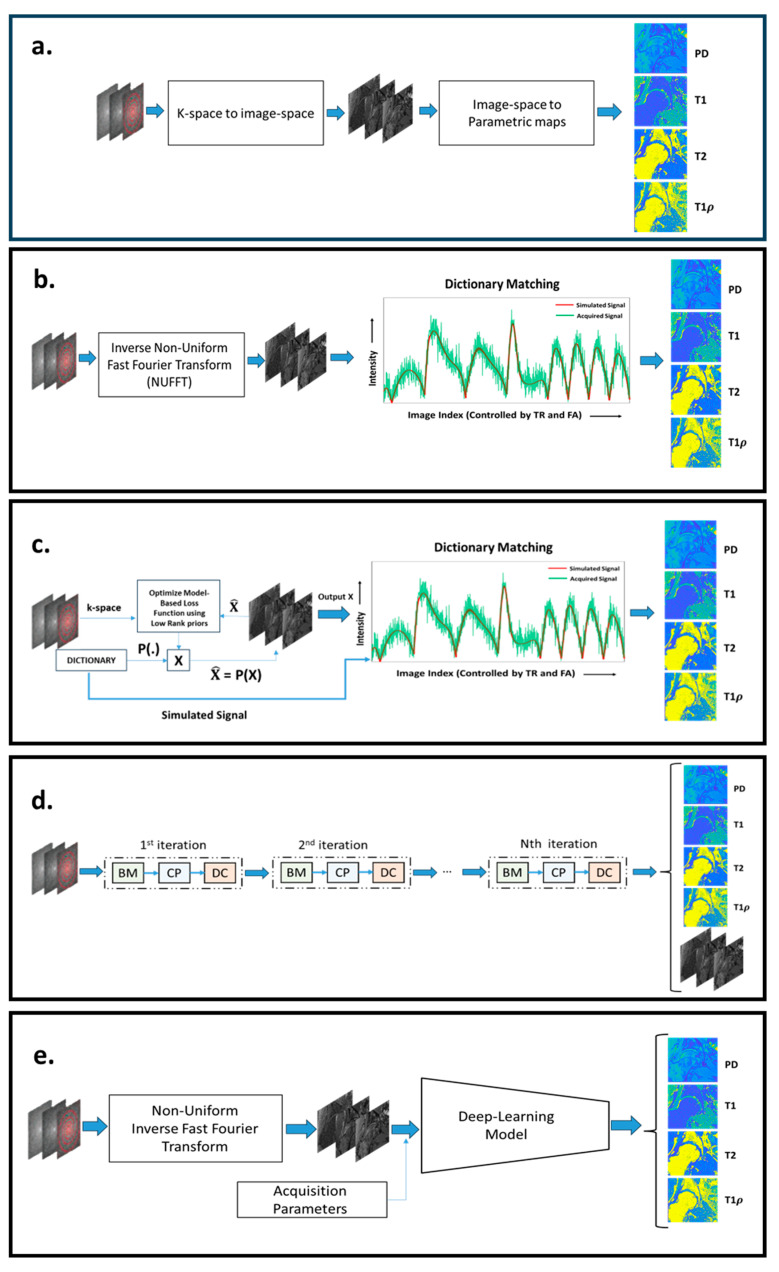
(**a**) The general configuration of different MRF reconstruction pipelines. (**b**) The conventional MRF reconstruction pipeline, with NUFFT used to reconstruct the image space and dictionary matching to recover the parametric maps. In (**c**), a model-based MRF reconstruction approach is shown, where the images are iteratively estimated from k-space, using image models, such as low-rank constraints. Dictionary matching is used to extract the parametric maps from the images. In (**d**), an unrolled network configuration for MRF reconstruction is shown, where an iterative-like structure is composed of a Bloch manifold projector module (BM), a learned decomposition module (CP), and a data-consistency module (DC). In (**e**), a mixed approach is shown, combining NUFFT to compute images and a deep learning network to produce parametric maps. The image was built from scratch but was inspired by [44,45].

**Figure 3 bioengineering-11-00236-f003:**
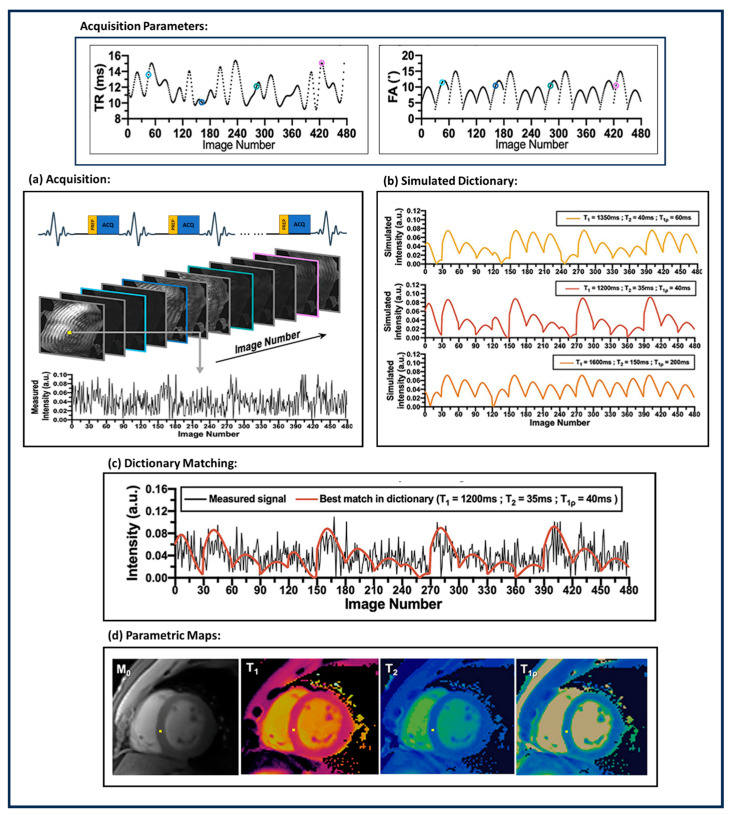
Workflow for cardiac MRF (cMRF). The workflow comprises (**a**) ECG-triggered MRF acquisition with motion corrected MRF image reconstruction; (**b**) simulating dictionaries corresponding to specific tissue properties by varying the acquisition parameters; (**c**) dictionary matching; and (**d**) the cardiac parametric map reconstructed after dictionary matching. The figure was derived from [15].

**Figure 4 bioengineering-11-00236-f004:**
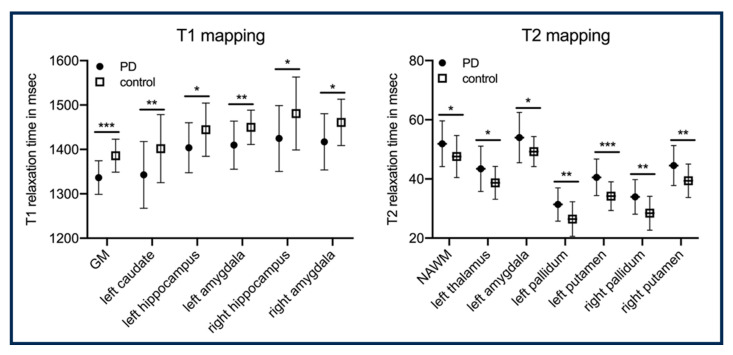
Illustrates significant differences in $T_{1}$ and $T_{2}$ relaxation times between group with Parkinson disease and control across different regions of the brain. NAWM: normal-appearing white matter. In this analysis there are 25 subjects per group. This figure is taken from [82]. *p*-values: * < 0.05; ** < 0.01; *** < 0.001.

**Figure 5 bioengineering-11-00236-f005:**
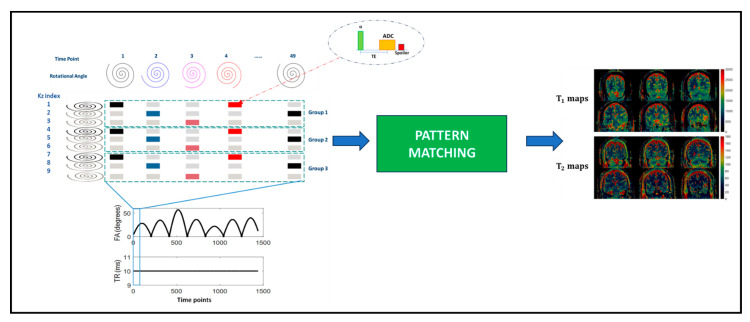
The 3D full-coverage brain acquisition with spiral trajectories. The k-space trajectories are interleaved and varied across the slice index to maximize k-space coverage. This figure is derived from [107].

**Figure 6 bioengineering-11-00236-f006:**
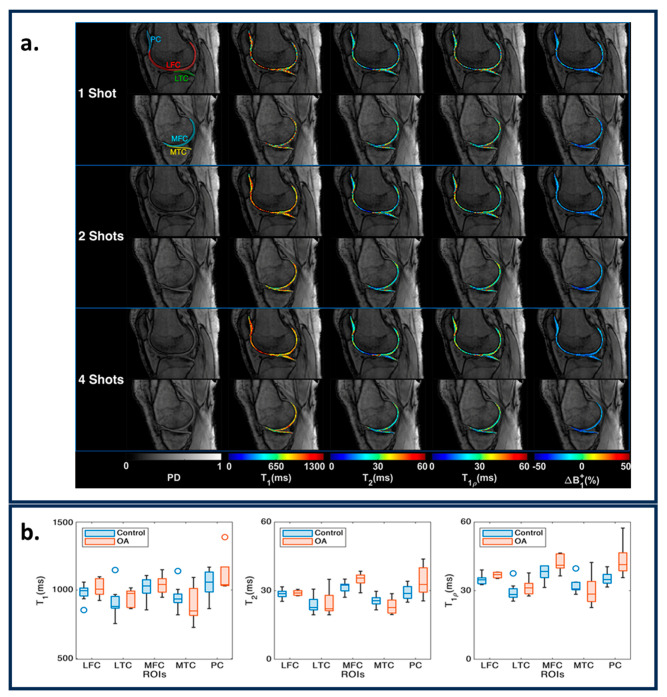
(**a**) Representative maps for one, two, and four shots in medial and lateral knee cartilages for PD images, $T_{1}$, $T_{2}$, $T_{1\rho }$, and ΔB1 + maps. The ROIs are shown in the shot one PD images. (**b**) The variation in parametric values for the knee between controls and OA subjects. The figures are derived from [86].

**Figure 7 bioengineering-11-00236-f007:**
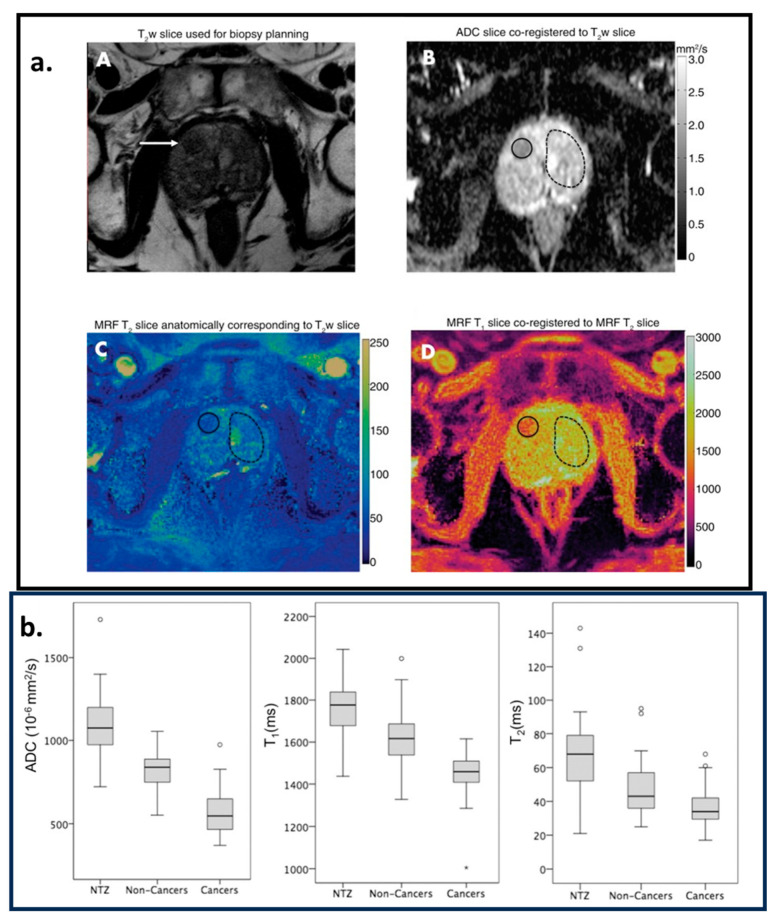
Region of interest (ROI) analysis. A cancer-suspicious lesion (white arrow) was identified via axial $T_{2}$-weighted ($T_{2}w$) acquisition, as shown in image (**A**). Image (**B**) is the apparent diffusion coefficient (ADC) map. (**C**,**D**) are the images corresponding to $T_{2}$ and $T_{1}$ parametric maps estimated from the MRF acquisition. (**A**–**D**) are coregistered. The solid circles in (**B**–**D**) correspond to the cancer-suspicious lesion. The dashed circles in (**B**–**D**) correspond to the visually Normal Transition Zone (NTZ). (**b**) Box-and-whisker plots for NTZ vs. non-cancerous lesions vs. cancerous legions for ADC, $T_{1}$, and $T_{2}$ parametric maps. This figure is adapted with permission from [110], Radiological Society of North America.

**Table 1 bioengineering-11-00236-t001:** Comparative analysis of preceding papers and present paper. The symbol ✓ indicates that the review paper covers trends in MRF pulse sequences, MRF pulse sequence optimization, dictionary matching, model based MRF reconstruction, Deep learning based MRF reconstruction and clinical application of MRF.

Ref.	Author	Year	Trends in MRF Pulse Sequences	Trends in MRF Pulse Sequence Optimization	Trends in MRF Dictionary Generation	Trends in Dictionary Matching	Trends in Model-Based MRF Reconstruction	Trends in Deep Learning-Based MRF Reconstruction	Application
[9]	Bipin Mehta	2019	✓		✓	✓			✓
[10]	Megan E Poorman	2020	✓		✓	✓			
[11]	Debra F. McGivney	2020		✓		✓	✓	✓	
[12]	Jean J. L. Hsieh	2020	✓			✓	✓		✓
[17]	Charit Tippareddy	2021		✓		✓			✓
Present Paper	Anmol Monga	2023	✓	✓	✓	✓	✓	✓	✓

**Table 2 bioengineering-11-00236-t002:** List of cited MRF papers by individual application.

REF	PARAMETRIC MAPS	CONTRIBUTION	APPLICATION
[75]	T_1_ and T_2_	Introduced a 3D cardiac MRF technique with respiratory motion compensation, enabling T_1_/T_2_ myocardial mapping in a single free-breathing scan	Cardiac
[76]	T_1_ and T_2_	Demonstrated the feasibility of generating multi-contrast synthetic Late Gadolinium Enhancement (LGE) images using MRF-derived post-contrast T_1_ and T_2_ maps	
[77]	T_1_, T_2_, M_0_, and FF	Applied an MRF approach to quantify water- and fat-specific T_1_ and T_2_, M_0_ estimation, and Fat Fraction (FF) maps for cardiac imaging in a single breath-hold exam	
[78]	T_1_, T_2_, and T_1ρ_	Proposed a 2D MRF method for cardiac imaging, offering simultaneous T_1_, T_2_, and T_1ρ_ mapping using an ECG-triggered GRE sequence with inversion recovery pulses	
[79]	$T_{2}^{*}$ and T_1_ maps	Utilized an MRF technique to simultaneously estimate perfusion, diffusion, T_2_*, and T_1_ maps	Brain
[80]	Water T_1_ relaxation (T_1_w) map Water T_2_ relaxation (T_2_w) map Amide exchange rate (ksw) map Amide volume fraction (fs) map Semi-solid exchange rate (kssw) map Semi-solid volume fraction (fss) map	Applied CEST-MRF with EPI readout and DRONE deep learning reconstruction for accurate brain tumor quantification	
[81]	CBF, BAT, T_1_, and B_1_^+^	Identified the crucial impact of TR patterns on MRF-ASL data, highlighting a sinusoidal pattern with a 125 TR period as the most consistently effective for spatial estimation	
[82]	T_1_ and T_2_	Demonstrated the capability of MRF-derived T_1_ and T_2_ maps in accurately identifying Parkinson’s disease and assessing disease severity	
[83]	Perfusion, CBVa, BAT, MTR, and T_1_	Optimized ASL labeling durations using the Cramer–Rao Lower Bound to enhance MRF-ASL signal sensitivity for brain hemodynamic quantification	
[84]	T_1_ and T_2_	Presented a 3D spiral projection acquisition with various interleaving spirals to increase robustness to rigid motion, revealing a significant improvement in motion-corrected quantitative maps compared to the motionless reference	
[85]	T_1_, T_2_, PD, and sodium density	Introduced a simultaneous 2D imaging method for proton T_1_, T_2_, proton density, and sodium density, utilizing a golden-angle radial trajectory without adversely affecting image quality for both protons and sodium	
[86]	T_1_, T_2_, T_1ρ_, and B_1_^+^	Developed a fast 3D-MRF technique for simultaneous T_1_, T_2_, and T_1ρ_ mapping in knee cartilage, revealing elevated T_1ρ_ in mild osteoarthritis with excellent repeatability	Musculoskeletal system
[87]	T_1_ and T_2_	A UTE-based MRF sequence was implemented to quantify T_1_ and T_2_ for muscle, bone, ligaments, and tendons	
[88]	PD, T_1_, T_2_, and T_1ρ_	Implemented an MRF sequence demonstrating rapid, simultaneous estimation of accurate PD, T_1_, T_2_, and T_1ρ_ maps of the lower leg muscle	
[89]	T_1_ and T_2_	An MRF acquisition is proposed that measures T_1_ and T_2_, which minimizes biases introduced by fat	
[90]	FF, off-resonance, B_1_^+^, and T_1_	Introduced an MRF approach, DBFW, utilizing RF spoiling to estimate T_1_ in water, T_1_ in fat, and FF maps, and it has four times faster acquisition than three-echo DIXON MRF	
[91]	T_1_, T_2_, and T_1ρ_	Demonstrated the feasibility of MRF for simultaneous bilateral mapping of T_1_, T_2_, and T_1ρ_ in the hip joint	
[92]	PD, T_1_, and T_2_	Presented an MRF technique that facilitates simultaneous measurement of PD, T_1_, and T_2_ maps for six radial hip sections	
[93]	T_1_ and T_2_	Assessed the use of the 3D MRF technique for simultaneous T_1_ and T_2_ mapping in breast tissues	Breast tissues
[94]	T_1_ and T_2_	Introduced three-point Dixon water–fat separation within the spiral MRF framework, enabling correction of fat-blurring using the CPR technique	
[95]	T_1_ and T_2_	Introduced a 3D abdominal MRF technique using a gated pilot tone (PT) navigator for simultaneous quantification of T_1_ and T_2_ without breath holding	Abdominal
[96]	T_1_ and T_2_	Demonstrated the feasibility of free-breathing pancreatic MRF at 1.5T and 3T, employing a spiral acquisition with oversampling of the center of k-space to mitigate in-plane motion artifacts	
[97]	T_1_ and $T_{2}^{*}$	Two-dimensional MRF sequence based on echo planar imaging was proposed using variable flip angles, TEs, and TRs, along with non-selective inversion pulses	Kidneys
[98]	T_1_, T_2_, T_1ρ_, and FF	Applied a gradient-echo liver MRF technique to enable simultaneous and comprehensive mapping of T_1_, T_2_, T_1ρ_, and FF in a single breath-hold scan	Liver tissue
[99]	T_1_ and T_2_	Verified the feasibility of T_1_ and T_2_ MRF for improving ovarian tumor detection, highlighting quantification capability without assessing its impact on detection compared to qualitative imaging	Ovaries
[100]	T_1_ and T_2_	Employed MRF to rapidly quantify relaxation times in the human eye at 7 T, achieving significant scan time reduction while maintaining detailed parameter maps without visible loss	Human eye

**Table 3 bioengineering-11-00236-t003:** Comparison of k-space trajectories typically used in MRF.

k-Space Trajectory	Advantages	Disadvantages
Cartesian	Easy and fast to reconstruct, with compatibility across all scanners	Inefficient k-space coverage and longer scan time, increased sensitivity to motion and susceptibility artifacts, and rapid switching of gradient coil can cause gradient heating and noise
Radial	Well-controlled linear trajectories, constant gradients, and fast readouts	Less k-space coverage and susceptible to gradient moment artifacts
Spiral	Better k-space coverage, short TEs, and relatively fast readouts	Susceptible to Eddy currents, with less precise trajectories
Rosette	$\mathrm{Better}\;\mathrm{k-}\mathrm{space}\;\mathrm{coverage}\;\mathrm{and}\;\mathrm{sampling}\;\mathrm{efficiency}\;\mathrm{compared}\;\mathrm{to}\;\mathrm{spiral}\;\mathrm{trajectories},\;\mathrm{improved}\;\mathrm{spectral}\;\mathrm{selectivity},\;\mathrm{and}\;\mathrm{can}\;\mathrm{be}\;\mathrm{used}\;\mathrm{for}\;\mathrm{fat}–\mathrm{water}\;\mathrm{separation}\;\mathrm{and}\;T_{2}^{*}$ mapping	The implementation of Rosette trajectories is computationally complex, and limited compatibility and hardware constraints are observed in older MRI scanners
Echo Planar Imaging (EPI)	Better k-space coverage compared to Cartesian acquisition	Long readouts and TEs, usually used for lower-resolution acquisitions

## Data Availability

No new data were created or analyzed in this study. Data sharing is not applicable to this article.

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
