# Peer review of "Emerging Trends in Magnetic Resonance Fingerprinting for Quantitative Biomedical Imaging Applications: A Review"

_bioengineering, 2024, doi:10.3390/bioengineering11030236_

Round 1

Reviewer 1 Report

Comments and Suggestions for Authors

This is a nice review about MRF and I only have two comments. 

1) Please emphase outstanding benefits using MRF against other fast acquisition techniques, e.g. compressed sensing. 

2) In outlook, how would the MRF be used at ultra-high field or ultra-low field?

Author Response

Manuscript ID-Bioengineering-2819334

We extend our sincere gratitude to the esteemed reviewers for their invaluable suggestions, dedicated efforts, and positive constructive comments. Your expertise and thoughtful insights have significantly enriched our manuscript, elevating its overall scientific quality. Your constructive feedback is deeply appreciated.

Reviewer 1

R1 comment1: Please emphasize the outstanding benefits using MRF against other fast acquisition techniques, e.g. compressed sensing.

Response: Thank you for the comment. We have now modified the text in the introduction between lines 56-59 to emphasize the benefits of MRF against fast acquisition techniques.

R1 comment2: In outlook, how would the MRF be used at ultra-high field or ultra-low field?

Response: Thank you for the comment, we have now added a paragraph reviewing trends in both ultra-high field and low field MRF in section 7 between lines 682-692.

Reviewer 2 Report

Comments and Suggestions for Authors

This manuscript is a comprehensive review focusing on current and future developments in magnetic resonance fingerprinting (MRF) for quantitative biomedical imaging. While largely complete, a few issues might be considered:

1. There are a number of existing recent reviews on MRF, e.g.:

Hsieh, J. J., & Svalbe, I. (2020). Magnetic resonance fingerprinting: from evolution to clinical applications. Journal of Medical Radiation Sciences, 67(4), 333-344.

Tippareddy, C., Zhao, W., Sunshine, J. L., Griswold, M., Ma, D., & Badve, C. (2021). Magnetic resonance fingerprinting: an overview. European Journal of Nuclear Medicine and Molecular Imaging, 48(13), 4189-4200.

Poorman, M. E., Martin, M. N., Ma, D., McGivney, D. F., Gulani, V., Griswold, M. A., & Keenan, K. E. (2020). Magnetic resonance fingerprinting Part 1: Potential uses, current challenges, and recommendations. Journal of Magnetic Resonance Imaging, 51(3), 675-692.

Kiselev, V. G., Körzdörfer, G., & Gall, P. (2021). Toward quantification: microstructure and magnetic resonance fingerprinting. Investigative Radiology, 56(1), 1-9.

etc.

As such, it might be briefly justified as how this review contributes a perspective that has not been addressed by prior work.

2. In the Introduction section, it is claimed that "Traditional quantitative MRI approaches... usually acquire a single quantitative imaging parameter at a time". However, it seems that if the MRI images are saved, additional quantitative imaging parameters could then be computed from the saved MRI image data, without the need to re-acquire the MRI images. This point might thus be clarified.

3. On the MRFs, it is then stated that "...this was done by comparing the measured signal against a set of pre-computed ideal signal evolutions (their fingerprints) from different pre-determined tissue types. The collection of fingerprints is also called a dictionary, as depicted in block (C) of Figure 1". For the general audience, this description might be expanded upon, especially as it pertains to the major focus of the review (i.e. MRFs): what is the "ideal signal evolution" for a tissue type/class? Would some tissue type be appropriately represented as a single fingerprint (or even a small set of fingerprints), especially if it is a broad group (i.e. for a "normal" type MRF, there may be numerous demographic/clinical factors such as age, ethnicity, gender, etc. that might not be reflected in the data used to construct the MRF)? All these considerations for constructing MRFs and their dictionaries/models (usually for reconstruction), might thus be explained further.

4. In Section 5.1, a brief overview/discussion of the distance metrics employed for dictionary matching might be warranted.

5. In Section 5.3, it is stated that "The deep learning model replaces the dictionary matching step". Exactly which parts of the "dictionary matching step(s)" are replaced, might be briefly explained.

Comments on the Quality of English Language

There are a number of minor grammatical/phrasing issues throughout the text, e.g.:

(Line 115) "...from both, technical advancements and application domains" -> remove comma after "both".

(Line 120) "plays the core role" -> "plays a core role"

(Line 121) "exciting, and sensing" -> "exciting and sensing"

(Line 134) "the paper [18]" -> might refer to the paper by specific author name; this might also be considered for following references of "the authors..."

(Line 376) "All that can..." -> "All these can"

(Table 1) "4> faster acquisition..." for [87] might be checked.

(Line 445) "motion artifact hence a..." -> "motion artifacts, hence a"

(Line 635) "Both, T_1 and T_2 values..." -> "Both T_1 and T_2 values"

In general, the manuscript might be carefully proofread.

Author Response

Manuscript ID-Bioengineering-2819334

We extend our sincere gratitude to the esteemed reviewers for their invaluable suggestions, dedicated efforts, and positive constructive comments. Your expertise and thoughtful insights have significantly enriched our manuscript, elevating its overall scientific quality. Your constructive feedback is deeply appreciated.

Reviewer 2

R2 comment1: There are several existing recent reviews on MRF, e.g.:

Hsieh, J. J., & Svalbe, I. (2020). Magnetic resonance fingerprinting: from evolution to clinical applications. Journal of Medical Radiation Sciences, 67(4), 333-344.

Tippareddy, C., Zhao, W., Sunshine, J. L., Griswold, M., Ma, D., & Badve, C. (2021). Magnetic resonance fingerprinting: an overview. European Journal of Nuclear Medicine and Molecular Imaging, 48(13), 4189-4200.

Poorman, M. E., Martin, M. N., Ma, D., McGivney, D. F., Gulani, V., Griswold, M. A., & Keenan, K. E. (2020). Magnetic resonance fingerprinting Part 1: Potential uses, current challenges, and recommendations. Journal of Magnetic Resonance Imaging, 51(3), 675-692.

Kiselev, V. G., Körzdörfer, G., & Gall, P. (2021). Toward quantification: microstructure and magnetic resonance fingerprinting. Investigative Radiology, 56(1), 1-9.

etc.

As such, it might be briefly justified as to how this review contributes a perspective that has not been addressed by prior work.

Response: Thanks for your comment. We have now added a new Table 1 in the contribution section, we have highlighted the contribution of the previous review papers and compared them against this review paper. Between Lines 124-136, we highlight the contribution of this review paper. Additionally, we also discuss the advantages and disadvantages of this review paper.

R2 comment2: In the Introduction section, it is claimed that "Traditional quantitative MRI approaches... usually acquire a single quantitative imaging parameter at a time". However, it seems that if the MRI images are saved, additional quantitative imaging parameters could then be computed from the saved MRI image data, without the need to re-acquire the MRI images. This point might thus be clarified.

Response: Thanks for the comment. I believe that the wording of the section didn’t clearly explain the issues associated with traditional quantitative MRI.  Hence, we modified the introduction section between lines 47-54 to clarify this critical point in the revised manuscript.

R2 comment3: On the MRFs, it is then stated that "...this was done by comparing the measured signal against a set of pre-computed ideal signal evolutions (their fingerprints) from different pre-determined tissue types. The collection of fingerprints is also called a dictionary, as depicted in block (C) of Figure 1". For the general audience, this description might be expanded upon, especially as it pertains to the major focus of the review (i.e. MRFs): what is the "ideal signal evolution" for a tissue type/class? Would some tissue type be appropriately represented as a single fingerprint (or even a small set of fingerprints), especially if it is a broad group (i.e. for a "normal" type MRF, there may be numerous demographic/clinical factors such as age, ethnicity, gender, etc. that might not be reflected in the data used to construct the MRF)? All these considerations for constructing MRFs and their dictionaries/models (usually for reconstruction), might thus be explained further.

Response: Thanks for raising this point. This is, in fact, a deeper point to discuss. We modify the text in the introduction between lines 81-86 to avoid this discussion there. We have included more details in the discussion section between lines 303-305.

R2 comment4: In Section 5.1, a brief overview/discussion of the distance metrics employed for dictionary matching might be warranted.

Response: Thanks for the comment. A brief overview of distance metric and the latest trends are  added in section 5.1 between lines 320-335

R2 Comment5: In Section 5.3, it is stated that "The deep learning model replaces the dictionary matching step". Exactly which parts of the "dictionary matching step(s)" are replaced, might be briefly explained.

Response: Thank you for raising this comment. Now, we have reorganized Section 5.3 in the revised manuscript to make a distinction between various applications of deep learning models in MRF reconstruction.

Reviewer 3 Report

Comments and Suggestions for Authors

In your paper you aim to review the best practices in each key aspect of MRF, as well as for different applications such as cardiac, brain, and musculoskeletal, among others. You might add other application specific  reviews to complete your Review, like: Cancers 2021,13, 4742. https://doi.org/10.3390/cancers13194742
or Curr Cardiol Rep. 2019 Jul 27;21(9):91. doi: 10.1007/s11886-019-1181-1.

Reproducibility is indeed an important aspect for quantitative imaging. The reproducibility of MRF has been thoroughly validated (among other Jiang et al.) Nevertheless only a small proportion of MRF features exhibits excellent repeatability and reproducibility, highlighting the importance of reliable MRF feature selection. It would boost the significance of your review if you could add more discussion on reliability and how to test it for the different frameworks of MRF.  Up to now most of the methods described on MRF were heuristically optimized. Optimization of acquisition and sequence parameters is therefore an open question and active area of research with multiple groups focusing on the problem. Maybe you can stress more words on how statistically learning algorithms like artificial intelligence can help to develop the field in that direction.

Author Response

Manuscript ID-Bioengineering-2819334

We extend our sincere gratitude to the esteemed reviewers for their invaluable suggestions, dedicated efforts, and positive constructive comments. Your expertise and thoughtful insights have significantly enriched our manuscript, elevating its overall scientific quality. Your constructive feedback is deeply appreciated.

R3 comment1: In your paper you aim to review the best practices in each key aspect of MRF, as well as for different applications such as cardiac, brain, and musculoskeletal, among others. You might add other application specific reviews to complete your Review, like: Cancers 2021,13, 4742. https://doi.org/10.3390/cancers13194742
or Curr Cardiol Rep. 2019 Jul 27;21(9):91. doi: 10.1007/s11886-019-1181-1.

Response: We have now added additional review papers for different applications in section 1.2 between lines 103-118 as suggested by the reviewer

R3 comment2: Reproducibility is indeed an important aspect for quantitative imaging. The reproducibility of MRF has been thoroughly validated (among other Jiang et al.) Nevertheless, only a small proportion of MRF features exhibits excellent repeatability and reproducibility, highlighting the importance of reliable MRF feature selection. It would boost the significance of your review if you could add more discussion on reliability and how to test it for the different frameworks of MRF. 

Response: Thank you for the comment. We agree with the reviewer. We have now expanded on variation in repeatability and reproducibility in the brain and prostate on MRF, between lines 693-722.

R3 comment3: Up to now most of the methods described on MRF were heuristically optimized. Optimization of acquisition and sequence parameters is therefore an open question and active area of research with multiple groups focusing on the problem. Maybe you can stress more words on how statistically learning algorithms like artificial intelligence can help to develop the field in that direction.

Response: We agree with the reviewer. We have now included more discussion on the topic in section 7 between lines 671-674.